# Analyzing Natural Digital Information in the Context of Market Research

**Evgenii Konnikov [1],\*, Olga Konnikova [2], Dmitriy Rodionov [1] and Oksana Yuldasheva [2]**

[1] Graduate School of Industrial Economics, Peter the Great St. Petersburg Polytechnic University, 195251 Saint Petersburg, Russia; drodionov@spbstu.ru

[2] Department of Marketing, St. Petersburg State University of Economics, 191023 Saint Petersburg, Russia; olga.a.konnikova@gmail.com (O.K.); yuldasheva.o@unecon.ru (O.Y.)

\* Correspondence: konnikov_ea@spbstu.ru; Tel.: +7-961-808-4582

**Abstract:** The dynamics of irreversible multidimensional digitalization of production and consumption processes can be described today with a linear-positive or even exponential function. A significant part of the information background of a product, enterprise or brand is formed by their consumers, competitors or partners on the Internet, which considerably increases its accessibility and spread. Such kind of information can be called natural digital information (NDI). Its high market value is counterbalanced by its inhomogeneity and complexity for analysis. The solution to this problem lies in the field of creating automated tools for its subsequent search, aggregation, primary processing, quantification and analysis. The aim of this study is to describe the unique methodological properties of market research based on natural digital information. In order to achieve this aim, this study analyzes the theoretical basis in the field of NDI research, defines the categories of NDI and sources of its formation, describes the key properties of NDI, determines its advantages in comparison with other types of market information, and suggests a basic methodology for conducting typical NDI-based market research. An applied research study was carried out according to the designed methodology to show its advantages, as well as to describe the unique methodological properties of market research based on processing of NDI. The main result of this study is a universal algorithmic model for analyzing NDI in the context of market research, which includes a mechanism for defining and categorizing the digital sources of NDI, a model for forming the key properties of NDI, and basic classes of NDI analytical metrics. The toolkit developed by the authors allows market research to be conducted without direct attraction of research subjects, which results in cost reduction and elimination of the phenomenon of social desirability; this creates the so-called reasoned advertising messages that meet the requests of the target audience, which is proved by the big data that underlie the presented methodology. The developed algorithmic model is universal for analyzing natural digital information, and, with minor adaptations, can be used by any subject conducting market research.

**Keywords:** natural digital information; market research; digital environment; information background; tokenization; tonality analysis; web-analytics



## 1. Introduction

Today, consumers regularly use the digital environment (social media, messengers, applications, blogs, etc.) to take part in discussions about the products and services of a company and express their preferences. Their buying decisions are being more and more affected by the recommendations obtained on the net. At present, 35% of consumers admit that they learn about new products via search engines, 27% via advertisements in social media, 25% via websites of manufacturing companies, 23% via recommendations and social media, and 22% via advertisements in mobile applications [1]. The number of users of the social platform Facebook amounts to about 2.5 billion people around the world (+13% by 2019); the second most popular is YouTube with 2 billion people (+33%

users a year); the third is WhatsApp with 1.6 billion (+23%) [1]. Expenses on internet advertising exceeded expenses on all other means of promotion as early as in 2016, and the total number of YouTube viewers exceeded the audiences of TV in 2018.

The year 2020 and the global coronavirus pandemic have spurred the development of the digital environment. The global expenses of consumers in mobile applications have grown by 10%. Almost half of developers claim that their earnings have risen by more than 20%. Even though the expenses of brands associated with digital advertising (including on mobiles) are reducing due to the coronavirus pandemic, the revenues of health and sports applications have gone up by 24%, while the incomes of financial services demonstrate an 18% growth, those of shopping applications by 15%, and those of food delivery applications by 10% [2].

Knowing this, many companies use the information generated in the digital environment to optimize their marketing and management decisions. The correctness of this approach was confirmed over the course of lots of academic research dedicated to this topic [3,4].

At the theoretical level, scholars have built models to measure the influence of information about the behavior of friends on social media on the decisions made by consumers [5]; to assess the interrelation between the information studied by the consumer on social media and dynamic pricing of new products [6,7]; to assess the impact of paid placement in search engines on the discount policy of a company [8]. As can be seen, most theoretical research is dedicated to analyzing the value of information in the digital environment for management decision making.

Even though this value was established theoretically for many business contexts, the opportunities that digital information provides for increasing the efficiency of company performance are not studied enough from the empirical perspective. Most empirical research studies in this field are dedicated to finding out the value of digital information for projecting various economic indicators in the spheres of finance, marketing, and information systems. For instance, in [9] the authors investigated the impact of universally accessible information on social media on improving the precision of daily sales forecasts. In [4] the authors suggested a prognostic model for the stock market using data from microblogs and demonstrated that the information on social media improves the projected return on the major financial indicators. In [10] the authors studied how sellers have to dynamically correct their marketing strategies to maximize incomes based on online reviews and comments. In [11] the author quantitatively measured the impact of restaurant reviews on social media websites on the attendance rates of restaurants and summarized areas for managing information on social media in this field.

Thus, attempts have been made at the empirical level to measure the influence of digital information on the performance indicators of a company and calculate its operational value. However, virtually all the above authors highlight some drawbacks of such kind of information. Primarily, it is the complexity of uniting two types of data (the internal operational data available for the company and the data from the digital environment to some of which the company has no direct access). Secondly, considerable amounts of information in the digital environment have an unstructured, text format, and this is not the type of data that researchers and practitioners traditionally work with. Some scholars suggest using methods such as natural language processing to code the feelings and emotions of commentaries, posts, and records on social media [9]. These methods have enormous potential for using digital information in completely new ways—primarily, in the field of market research.

The modern era of digitalization makes professionals working in the sphere of market research face a difficult choice. On the one hand, the popularity of once common methods of research, such as focus groups and surveys, is reducing. On the other hand, new sources of information spring up, giving big opportunities for analysis using the digital environment itself.

Today, methods such as web-analytics are becoming more and more popular, representing a system for acquiring and analyzing data on the visitors to websites and accounts on social media in the context of the characteristics of their behavior in the digital environment. The modern trend is that web-analytics is becoming simpler. This means that its results can be interpreted by people that do not have specialized education or skills, for example, using the Google Analytics app. Knowing which target actions the user took on a website or in an account, which pages were browsed and how much time was spent on each of them, and which queries or, maybe, hashtags were used to find the digital representations of the company, market researchers can analyze the specific needs and vocabulary of the target audience, improve and optimize websites and business accounts, and measure the efficiency of digital marketing actions (for example, analyzing such indicators as: contract price in case of advertising or price of visiting the website). The totality of these data is a classical array of big data, which have such characteristics as volume (meaning the size of physical volume), rate (meaning the growth rate) and diversity (meaning the capability of simultaneous processing of the most diverse information from various sources of origin) [12].

However, apart from web-analytics, there is a far more substantial type of information in the digital environment, which authors call natural digital information (hereinafter, NDI). Natural information is an unstructured array of data that implies processing by common mind. The main characteristic of natural information is that it emerges outside the context that it will be analyzed. As a matter of fact, it is metadata, which is not initially prepared for analysis. The main aim of this paper is to describe the unique methodological qualities of market research based on natural digital information processing. In order to reach this aim, the following objectives have to be accomplished:

1.  Analyzing the theoretical basis in the field of NDI research, defining the category of NDI and sources of its formation.
2.  Describing the key qualities of NDI, determining its advantages in comparison to other types of market information.
3.  Forming a basic methodology for conducting typical NDI-based market research.
4.  Conducting applied research in accordance with the devised methodology.
5.  Processing the research results, describing the unique methodological qualities of market research based on natural digital information processing.

## 2. Literature Review

The concept of natural information has been used in various sciences for quite a while, originally as an element of natural language [13], with natural information being opposed to precise, rigorous information [14].

According to the best-known studies dedicated to natural information [15–22], it is considered as "biologically realistic", being the result of the "epistemic contact" of man with the environment and corresponding to his values. In many cases, authors consider natural information as an environment in itself (both objective and context-dependent), helpful for the user (consumer) in various conditions where it occurs and is part of cause and historic prerequisites.

There is another point of view, according to which natural information always includes both the relations inside the external environment and the relations of a perceiving organism towards its environment [23]. In addition, natural information helps to maintain the objectivity of information relations and expands the capabilities of the subject perceiving the information, who uses these objective relations for their own specific purposes [24,25].

When considering natural information emerging in the digital environment, it can be divided into two categories:

1.  Universally accessible NDI, which include:
    a.  Information stipulated by the user in their profile (account) on various social media: gender, age, place of residence, interests, contacts, hobbies, etc.

b.　Messages written by the user in open forums, in open groups on social media as comments, answers to questions, reviews, posts in blogs, etc.

c.　Number of views, likes, reposts, comments to various messages (posts, notes, etc.) in the digital space.

2.　Private NDI, which include:

a.　Data of personal correspondences in messengers and messages on social media.

b.　The so-called "passive digital footprint" [26]: history of visits to websites, geolocation, data on purchases made using credit cards, etc. As a matter of fact, passive digital footprint, as an example of private natural digital information, is the closest to web-analytics data, and it is not always easy to draw the line between them.

The source of origin of natural digital information was technological innovations of the mid-2000s such as Web 2.0 technology, which resulted in the formation and booming of such a phenomenon as user-generated content (UGC) [27,28]. UGC is one of the many components of NDI. The main idea borrowed by the authors from the concept of UGC is that information emerging in the digital environment is not just an object of passive consumption, but that it is created, shared and consumed by people that are active users of the Net.

No doubt, natural digital information is of great interest to researchers, because it is commonly known from the perspective of consumer psychology and the theory of market research that the less a person is aware that they are an object of research, the more reliable the information that they share is, even if they do not suspect that [29,30]. This is due to the fact that research based on natural digital information (as opposed to traditional empirical methods of consumer research) helps to cope with such a property of consumer psychology as "social desirability", i.e., conscious or partially conscious tendency of the subject to give more socially approved, desirable answers in the course of research.

Some researchers investigated the impact of the convincing force of natural digital information generated from the communication of users on social media [31]. To be convincing for a reference group, this information, must have the following qualities:

1.　Timeliness.
2.　Precision.
3.　Depth.
4.　Relevance.
5.　Trustworthiness.

These conclusions rely on the theory of social media [31–33], theory of social penetration [31], social exchange theory [31,34] and the theory of rumor spread [35], since social media are often perceived as a digital version of oral communication [31], with only the distinction that every user can simultaneously communicate with hundreds or even thousands of peers, and in order to form such an institution as influencer marketing, each of these hundreds and thousands users is needed. Otherwise, the system would collapse, because it is not brands that communicate with consumers, but consumers communicate with each other about brands [36]. Information on social media affects buying behavior through two mechanisms: the "attention" effect, which fixates the awareness about products, and the "approval" effect, which informs users about the quality of a product based on the online comments of their friends and influencers [37–39].

Thus, the following advantages of NDI can be defined:

1.　NDI is more reliable in comparison to data obtained through "traditional" instruments of market research, such as focus groups, in-depth interviews and surveys. This advantage is caused by such a property of NDI as independence of obtaining: in many cases, users do not suspect that the digital information they leave behind is somehow collected and analyzed, which deprives it of such a property as "social desirability", mentioned above. This advantage helps absolutize such a property of information as "uncertainty reduction" [40] or "entropy production constraint" [41]. This makes

it possible to ideally refine natural digital information in the information hierarchy theory, according to which the purpose of using any information by the user can be characterized by the movement along the route: thermodynamic search→uncertainty reduction→appearance of meaning [42].

2.  The price of obtaining such information is minimum. Actually, most NDI is open access (on websites, forums, social media, etc.).

3.  The time necessary to collect NDI is also minimal and, most importantly, this process can be automated if a single updatable data frame is created to collect a pool of NDI determined in advance, with the frequency set by the researcher.

If the process of NDI acquisition is automated, it helps to solve the problem of timeliness, i.e., the necessary degree of up-to-datedness of the information in relation to the problem being tackled, because obsolete, not up-to-date information will be replaced with new up-to-date data in real time mode (which helps to implement such a property of an informed system as categorization of input data or sensations as signs of external objects [43]).

### 3. Materials and Methods

This paper considers in detail just one of the spheres of use of natural digital information, namely: analysis of NDI can contribute to creating effective advertisements (e.g., in the field of context and targeted advertising, integration by bloggers, etc.), which have a high degree of nativity, i.e., correspondence to the characteristics and interests of prospects and specific features of advertising platforms. Today, when the trend of digital detox (partial or complete rejection of using digital devices) is spreading and banner blindness (a phenomenon in web-usability when visitors to websites do not notice advertising blocks or objects resembling them) is common, reasoned design of advertising messages is the only way to transform the consumer behavior that the advertiser strives for.

Reasoned design is understood as working on three components of advertising messages through obtaining answers to the following questions:

1.  The visual (graphic) part of the advertisement: What color solutions should be chosen to attract maximum attention of the user? What fonts should be used? How bright or neutral should the appearance of the advertising post be?

2.  The text (content) part of the advertisement: What key words will make the consumer want to make the target actions? What amount of text is going to be the best for perception? What should be the tonality of the advertisement?

3.  The targeted part of the advertisement: What settings of the target audience should be chosen when the advertisement is created? Who will it be shown to, on what platforms and under what conditions?

Thus, natural digital information processing ensures the following possibilities:

*   Retrieving posts (on social media, forums, blogs, etc.) that caused the biggest responses of consumers (likes, reposts, comments, views) and their further visual and content analysis. What is more, not only text, but also photo/audio/and video messages can be analyzed.

*   Analyzing the tonality of text information, i.e., automated identification of emotionally colored vocabulary and emotional assessment (opinions) of the authors towards the objects discussed in the text. This analysis makes it possible to define the emotional qualities of the content and, consequently, form the most effective tone of the marketing appeal for its consumers, as well as to determine the emotional color of the information background, which, in turn, can have an impact on the behavior of various players.

*   Analyzing the meaningful content, aimed at identifying the lexical categories, "tokens", i.e., meaningful elements (words, phrases, symbols) of the text, and their quantified analysis via calculating the frequency of occurrence of the most common words and creating an importance matrix of tokens. Importance analysis can identify

the substantial features of the presented templates, and ignore any insignificant and, correspondingly, trivial inaccuracies they contain [14].

- Analyzing the social connections of the target audience, using a method of social graphs, which makes it possible to create substantiated avatars of the company's target audience (an avatar or a portrait of an ideal client is a profound, step by step description of an individual representative of the target audience), search for look-alike audiences, representing the result of the analysis of the initial database (for instance, analysis of real consumers of the company or actual subscribers of a company's community) in terms of the presence of resemblances (interests, behavior, other factors) and search for the maximally similar users among all registered accounts (as a rule, this is performed using neural networks).

NDI is treated using a procedure called parsing (a process of automated data acquisition with its further treatment and analysis). The program that carries out the collection and a syntactic analysis is called a parser. The process of parsing allows two different methods of information treatment to be accommodated, which may seem contradictory at first sight: a natural method of treatment (characterized by, for example, the way people read books and understand speech) and a method of machine treatment (computer-aided processing of programming languages) [44]. The method of "intellect modelling" [14] can be used to obtain very promising results, especially due to the possibility of "discrepancy correction". It is assumed that such a "resistant to error" interface can make the communication between a human and a machine more effective in future.

The main stages of NDI parsing can be characterized as follows:

1. Data search. The initial HTML code, for example, that of a website page is uploaded in a parser. A script, which breaks up the entire text into lexemes, starts to work with the code, highlighting the necessary information.
2. Information retrieval. Data are searched thanks to a certain collection of characters, describing the purpose of the search. This collection is also called regular expressions. They can be used to highlight only those fragments that are of interest in the entire array.
3. Data saving. After it is obtained, information is saved in the form of tables or is included in the database.

If specialized methods are applied, natural digital information can be transformed into an array of data suitable for quantitative processing (Figure 1).

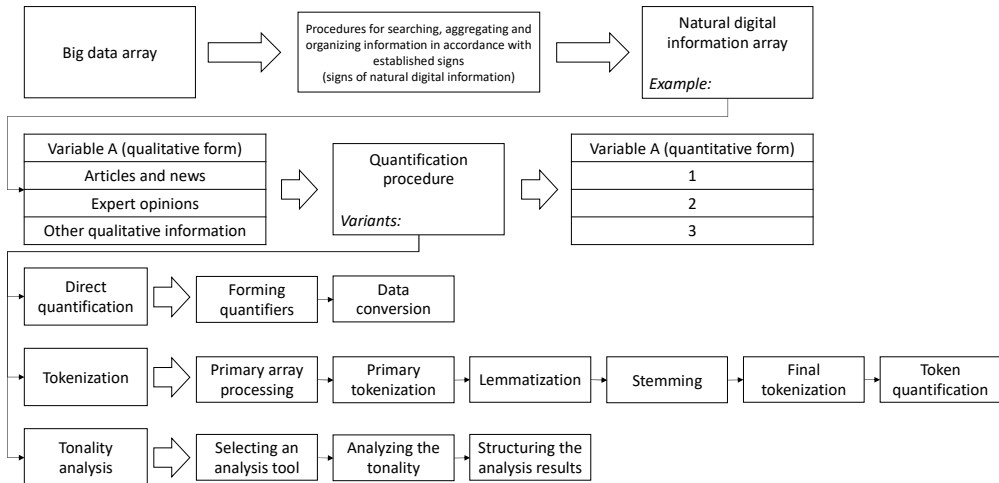

**Figure 1.** The process of processing of natural digital information.

As can be seen in Figure 1, after natural information is collected and categorized, it has to be quantified prior to the analysis procedure. It should also be noted that natural digital information is only part of the big data generated by humans in the information

environment. Its particular quality is that it is formulated in a natural form and requires decoding through human consciousness. This is what determines the need for its preliminary quantification using special tools. There are many quantification tools, but the most common ones are presented in Figure 1. Direct quantification is understood as conversion of the available nominal or ordinal lexemes into their alternative (numerical) form.

Then, the tokenization procedure mentioned above takes place—the text is broken down into meaningful elements (words, phrases, characters) called tokens. However, different forms of the word often have to be processed in the same way. For example, when searching according to queries "marketing" or "market researcher", the same answers are expected. Special procedures for processing tokens are used for this purpose, such as lemmatization and stemming. Lemmatization is changing the word to its dictionary form. Stemming is the process of finding the stem of the word, which does not necessarily coincide with the root of the word. It should be noted that the root of a word (especially when it comes to verbs and adjectives) can have a fundamentally different interpretation depending on the prefixes and suffixes. In this regard, in order to preserve the pledged meaning of the word, it is necessary to highlight its basis. Due to subsequent application of these procedures, a much more effective (precise) variable can be formed. In many tasks, using the most frequent words create noise. For example, in the case of a full text search, the system can return almost all documents, if the query contains prepositions. Thus, the most frequent words are filtered as "stop words" and are not used in analysis.

After the data have been collected and the data frame based on them has been formed, the researcher moves on to develop a methodology for data processing, which, as a matter of fact, means selecting tools for analyzing data and working out an algorithm for their processing.

As an example, the authors suggest considering a methodology for investigating natural digital information in order to form a reasoned design of an advertisement (based on the example of an advertisement for 2D animation online courses) and analyze the specific features and preferences of the target audience (see Results section). The goal in this case is very private, but the results will be the basis for forming a single methodology and tools for analyzing natural digital information in the context of market research.

## 4. Results

The implementation of the project denoted in the previous section can be described via the following sequence of steps:

*Step 1—Selecting communities for analysis and obtaining information about the subscribers of the communities*:

The subscribers of three 2D animation communities on the social media platform Vkontakte (Russia) were selected as the audience for parsing: "Motion Design and Animation", "Adobe After Effects" and "Volnitsa". When the communities for parsing were chosen, such criteria were taken into account as the correspondence to the subject (vector graphic animation), a relatively large number of subscribers (over 30,000), subscribers not hidden by the administrators, the communities being "alive" (posting is regular) and having an active audience (not only can the subscribers comment, but they can also make posts).

The Python programming language was used to create a simple parser and obtain a data frame with information about 6000 active subscribers of these communities (such as gender, age, city and marital status). The data frame analysis provided the following results:

- Males are more active in the analyzed communities in comparison with females (M—3344 people; F—2656 people).
- It is not common for users to indicate their marital status, so when the settings of the advertisement are chosen, this criterion is not to be considered.
- The target audience comprises people living in metropolises (Saint-Petersburg—2452 people; Moscow—1395 people).
- The average age of an active subscriber in the analyzed communities is 33.5 years.

*Step 2—Obtaining posts from the wall of the communities, as well as the posts from the subscribers' walls*:

Using the created parser, it was possible to obtain 1800 posts from the analyzed communities for the period from 2014 to 2020 with information that included the text of the post, the date of the post, the number of likes, reposts and views.

As for the parsing of posts and comments from the subscribers' walls, 20 posts and 7 comments to any post were viewed for each 6000 IDs (unique users), revealed at the previous step. This substage of the analysis was the least effective, which was due to there being very few comments under the subscribers' posts, a small quantity of posts themselves and a large number of closed pages.

*Step 3—Tokenization of posts*:

For tokening the posts and obtaining the least or the most frequent and most important tokens, the text has to go through several phases. The first one was the conversion of the list of texts into a line and removal of odd characters for further correct analysis. Using the *nltk* library for Python, the so-called stop words were removed (including various conjunctions, prepositions, etc.—everything that was not meaningful). The lemmatization procedure was used to change the words to their dictionary form. Then, in an experimental way, it was revealed that it would be better to tag the text and remove the unnecessary word classes, and then, carry out stemming—reducing the form of a word to a more general one, since stemming changed the words too much, tagging did not work properly afterward, and word classes were defined in an incorrect way. The outcome was the retrieval of the most frequent tokens.

Then, the procedure of comparing the tokens from each post and general selection of tokens was carried out. The text of each post went through the same phases (apart from tagging) as the general text, but afterward, its tokens were compared to the general ones and brought to a separate list, which was later added to the data frame. Finally, the general data frame was obtained with the tonality and tokens of each post, which was added to the initial data frame. It was used to identify the tokens in the best posts by any parameters (for example, by the number of likes or the number of reposts). Table 1 contains the most frequently occurring tokens in the 25% quartile of posts that are reposted most often.

**Table 1.** The most frequent tokens in 25% quartile of posts that are reposted most often.

| Token | Frequency of Occurrence |
| --- | --- |
| Animat | 21 |
| Work | 5 |
| Animat, lesson | 4 |
| School, program, animat, week, matter, lesson, privat, work, new, wish, project, topic | 3 |
| Creat, begin, program, animat, lesson, privat, work, author, result | 3 |

Analysis of Table 1 leads us to conclude that the biggest interest of active users of the analyzed communities is provoked by the description of lessons and projects, rather than, for example, visually attractive content and animations.

*Step 4—Tonality analysis of posts*:

A free of charge library for the Python programming language called *Dostoevsky* can be used as one of the tools for analyzing the tonality of the texts. It allows the following characteristics of text tonality to be considered: (1) skip is the level of text insignificance, (2) neutral is the level of text neutrality, (3) negative is the level of text negativity, (4) positive is the level of text positivity, (5) speech is the level of text naturalness (belonging to informal language). All the tonality characteristics in the library are measured from 0 to 1.

Within this research study, tonality analysis was performed for the cumulative data frame from the list of posts of the above communities, comments to them, as well as posts on the subscribers' walls. The analysis showed that, according to the average tonality, the

posts of the communities were more negative than positive (0.1 being positive, 0.68 being negative). Comments are also of a more negative than positive tone (however, the gap was smaller, with 0.08 being positive and 0.1 being negative), but the positivity and negativity of the subscribers' posts were expressed in approximately the same proportions (0.06 being positive and 0.04 being negative).

*Result—Creating reasoned drafts of context and targeted advertisements*

Targeted advertisement 1 in the news feed of social media Vkontakte:

- In order to advertise the online course, a picture has to be selected. It should meet the following identified criteria: represent vector graphics, demonstrate the process of animation, be executed in a style popular for vector graphics.
- The target audience should be the active subscribers of the analyzed communities, who are interested in motion design and want to develop in this field. They most often like and repost the recordings of video lessons and various tutorials, so the main emphasis in the advertisement should be placed on the availability of free video lessons.
- The time for demonstration of the advertisement should be set from 9 a.m. to 2 p.m., since this is the time when the subscribers most frequently repost the posts of the communities.
- The projected target audience of the advertisement is 6300 people, covering approximately 30% of the target audience, with the total budget being around RUB 700 (according to the statistics of the Vkontakte Advertising Account).

Context advertisement №2 was created using *Yandex Direct* (after Google Ads, the second most widely used platform for creating contextual advertisements in Russia):

- The time for demonstration of the advertisement was chosen according to the previously obtained data: the target audience is most active within the period from 9 a.m. to 2 p.m., and from 4 p.m. to 7 p.m.
- Since, according to the data obtained, it was found that most of the target audience consists of males, corrections were concerning consumers' gender and age: the average rate was raised by 100% for males aged 25–34, and by 50% for males 35–44. Thus, the priority of displays was increased for those living in Moscow and St. Petersburg, because it was already previously established that they comprise a major part of the target audience.
- The key phrases chosen for demonstration of the advertisement were "animation lessons", "animation school", "creating animation", and "motion learning". The key phrases were identified as a result of the tokens obtained from the most reposted posts of the communities.

Thus, the parsing and further analysis of the natural digital information about the users of the product helped to create several advertising messages to the target audience that were reasoned from graphic, content and targeting perspectives. It should be noted that we analyzed NDI exclusively in the Russian language because the practical case that we described was implemented in the Russian market. However, our proposed solutions and conclusions are universal for natural digital information presented in any language. Additionally, it should be noted that many languages have unique specifics, primarily related to the ways and lexical traditions of reflecting information in the text. This specificity should be taken into account, but its presence does not contradict our conclusions and proposed solutions.

## 5. Discussion

The conducted research allows us to conclude that sequential analysis of natural digital information can be used to solve important practical problems of modern marketing. The formed characteristics of context and targeted advertising are the consequence of automated collection, processing and analysis of NDI contained on social media and formed outside the context of realization that it can be used for market research. Thus, this information is very effective from the marketing perspective. However, due to its inhomogeneity,

inconsistency, non-structuredness and differentiation in form and content, it cannot be used within the existing classical methodology of market research. Thus, the methodology of market research based on processing exceptionally natural digital information implies unique properties, different from the classical methodology of market research.

The analysis of the results obtained in the applied research study brings us to the conclusion that market research based on NDI processing, first of all, implies the need for identifying the digital source. Digital sources of natural digital information are extremely differentiated, both in terms of space and time. Essentially, the following digital sources of NDI can be highlighted:

1.  Social communication portals. These portals imply free communication between representatives of society, outside commercial goal setting. The bases of these portals are social media and thematic forums. It should be noted that these portals can be differentiated by thematic principle, geographic principle, demographic principle, etc. Information formed in the context of interaction of the subjects of these portals is the most significant for market research, since it is formed outside goal setting, which can reduce the representativeness of research results.

2.  News portals. These portals imply the formation of exceptionally targeted news content. This content is also natural digital information. However, its use in market research implies a specific totality of goals and objectives. This content has one-way mass impact on the representatives of society, thus forming a single information environment. If this content is quantified, the state of the information environment can be analyzed and its influence on society can be characterized.

3.  Commercial portals. The information on these portals implies exceptional commercial goal setting. In the first stage, these are the online stores and information websites of some enterprises. If the information presented on these portals is analyzed, the targeted impact of the subject on its consumers can be characterized, and, therefore, the efficiency of this impact can be determined through analysis of business activity results.

4.  Ideological portals. The information presented in these portals implies goal setting different from commercial. However, the structure of the impact made by this information on consumers is identical to commercial portals. We herein mean information portals, thematic websites, websites of religious and political organizations, etc. Thus, analysis of this information is virtually identical to the analysis of information on commercial portals. However, the effectiveness of these portals is not measured by commercial metrics.

The presented classification can, of course, be expanded. However, the highlighted groups are sufficient for differentiating the sources in terms of market research. A source is selected primarily depending on the goals and objectives of the market research that is carried out. It should also be noted that different sources of natural digital information can be used in one research study.

Following the results of the identified source, the properties of the natural digital information in it also have to be characterized. Natural digital information can be characterized in accordance with descriptions of the following properties:

1.  Content properties. These properties are determined by meaningful and lexical content. The range of vocabulary that characterizes the tools for presenting the information, and the themes, determining the scope of interests of the information carrier, are valuable to the researcher. Structuring and analysis of the content properties of natural digital information help the market researcher answer the questions of "what" mediates the mind of the user, competitor, or any other object of research and "what vocabulary apparatus" the object of research uses in narration, and, therefore, what is the most effective way to interact with them.

2.  Tonality properties. These properties define the emotional tone of the content. The tonality analysis tools existing today can be used to describe such characteristics of natural digital information as the level of positivity, negativity, neutrality and

many others. The assessment and analysis of the tonality properties of NDI help to, primarily, determine the relation of the research object to the content, as well as to conclude about the tonality properties of a potentially effective marketing message. Tonality properties are secondary towards the content ones, but they are the ones which determine the emotional vector of the marketing message in relation to the content.

3. Metaproperties. These properties define the time and place of generation of natural digital information as well as demographic, geographic, psychological, psychographic and other properties of the generation source. The description and analysis of these properties make it possible, first, to categorize the objects of market research, to identify the dependences between individual properties of objects and the tonality and/or content properties of the generated information, etc. These properties are determinant towards the content properties and can form a binary array of primary information. Its statistical analysis provides market researchers with a totality of comparative assessments, based on which meaningful conclusions can be made.

4. Context properties. These properties help to describe the context within which the content is formed. Natural digital information can be contextually differentiated in accordance with the following attributes:

    a. Formation level. The formation level is understood as a characteristic of the cause-and-effect chain as a result of which the content was formed. According to this attribute, it is possible to distinguish primary content (natural digital information formed due to reasons detached from it), secondary content (natural digital information formed as a reaction to primary content (treatment of the primary content is the cause, the secondary content is the effect)), high-level content (natural digital information formed as a reaction to the secondary or other high-level content).

    b. Content of single information environment. Any natural information is formed because of the single information content. Differentiating one or other content element (political instability, war time, period of mass unrest, sport achievements of the nations, etc.) helps to correct the obtained analytical results and determine their importance for some other period.

    c. Tonality of the single information environment. As noted above, natural digital information is formed under the effect of the single information context, which determines the influence of the tonality of the single information context on the properties of natural digital information. Considering the tonality of the single information environment allows the universality of the obtained analytical results to be defined.

Comprehensive description of the differentiated properties of natural digital information allows the researcher to form a single array of analytical data, which, if studied and compared, will help to reach multi-dimensional conclusions and answer the questions asked according to the goal setting of the market research. For each of the distinguished properties, a unique totality of metrics is defined based on the tools of applied statistics. The following can be highlighted as the basis classes of these metrics:

1. Connection metrics. This class includes tools reflecting the strength and nature of connection between the quantitative expression of properties of various totalities of natural digital information, on the one hand, and the properties of the analyzed array of natural digital information and the properties of the single information background, on the other hand. The tool basis for developing such metrics includes various types of correlation coefficients and regression coefficients.

2. Comparability metrics. This class has tools reflecting the relation between quantitative characteristics of the properties of various totalities of natural digital information as well as the properties of the analyzed array of natural digital information and the properties of the single information background. The tool basis for developing such

metrics includes comparison measures, dispersion indicators, as well as procedures for comparing dynamic and spatial coefficients.

3. Dominance metrics. This class includes tools to determine the most significant quantitative characteristics of natural digital information as well as the most significant representatives of an array of natural digital information. All tools in this class are based on distinguishing objects with maximal or minimal quantitative values of their characteristics.

4. Exceptionality metrics. This class includes tools allowing us to identify statistical outliers and unreliable values both among quantitative characteristics of natural digital information and in a totality of consistent objects of an array of natural digital information. Such metrics are applied to increase the quality of the analyzed array, to improve the reliability of the research results, and to substantiate the significance of the analytical results obtained.

Table 2 provides examples of metrics to quantify and analyze the properties of NDI. The examples given in the pivot table are not exclusive, but from a presentation point of view, we decided to demonstrate the described metrics for the following base values:

- $t_{p_i}$—the level of positivity of post *i*.
- $\overline{t_p}$—the average level of positivity of all posts in sample *n*.
- $N_{l_i}$—number of likes for post *i*.
- $\overline{N_l}$—the average number of likes for all posts in sample *n*.

**Table 2.** Examples of metrics to quantify and analyze the properties of NDI.

| Type of Metrics | Aim of Using | Indicators | Example |
|---|---|---|---|
| Connection metrics | Reflect the strength and nature of connection between: <br> - the quantitative expression of properties of various totalities of natural digital information <br> - the properties of the analyzed array of natural digital information and the properties of the single information background | correlation coefficients <br> regression coefficients | $I_{con.} = \dfrac{\sum_{i=1}^{n}\left(t_{p_i}-\overline{t_p}\right)*\left(N_{l_i}-\overline{N_l}\right)}{\sqrt{\sum_{i=1}^{n}\left(t_{p_i}-\overline{t_p}\right)^2}*\sqrt{\sum_{i=1}^{n}\left(N_{l_i}-\overline{N_l}\right)^2}}$ |
| Comparability metrics. | Reflect the relation between: <br> - quantitative characteristics of the properties of various totalities of natural digital information <br> - the properties of the analyzed array of natural digital information and the properties of the single information background | comparison measures <br> dispersion indicators <br> comparing dynamic and spatial coefficients | $I_{com.} = \sqrt{\dfrac{\sum_{i=1}^{n}\left(t_{p_i}-\overline{t_p}\right)^2}{n-1}}$ |
| Dominance metrics | Determine: <br> - the most significant quantitative characteristics of natural digital information <br> - the most significant representatives of an array of natural digital information | maximal or minimal quantitative values | $I_{dom.} = \max_{t_p}(i_1 \ldots i_n)$ |
| Exceptionality metrics | Identify statistical outliers and unreliable values: <br> - among quantitative characteristics of natural digital information <br> - in a totality of consistent objects of an array of natural digital information | identifiers of statistical outliers | $I_{dom.} = \max_{N_l}(i_1 \ldots i_n)$ |

According to the results of the analyzed metric values, the researcher can reach the necessary marketing conclusions and achieve the analytical objectives of the research. The presented totality of unique methodological properties of the market research based on natural digital information processing helps to form a relevant algorithmic model (Figure 2).

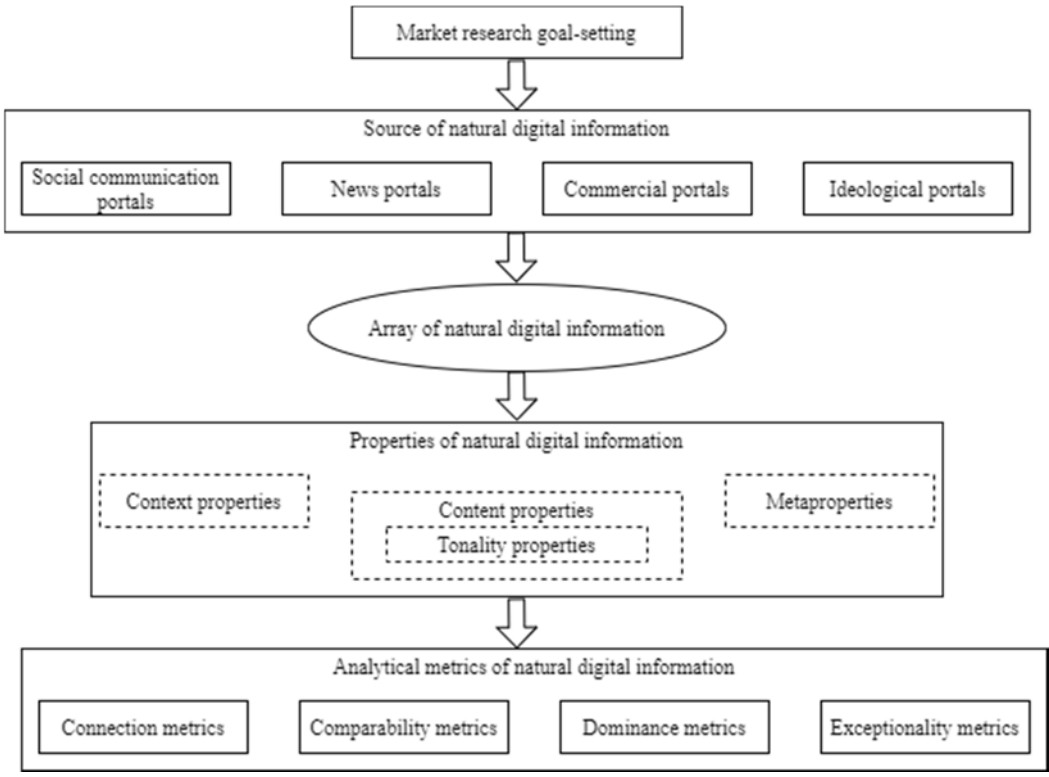

**Figure 2.** Algorithmic model for analyzing natural digital information in the context of market research.

The model obtained has a universal character and can be used in market research with various goal setting. As can be seen in Figure 2, this model considers the sequence of the process implemented to analyze natural digital information and the cause-and-effect nature of formation of its qualities.

## 6. Conclusions

This paper discusses the results of the studied properties of natural digital information in the context of market research. According to the study, at the time of the irreversible process of economy digitalization, natural digital information is becoming a more and more significant analytical resource, in particular, for market research. The quality of formation of natural digital information determines the value of its content for a market researcher, first, because it is formed outside the context of realizing the possibility of its analysis. However, the form of presentation of this information is less universalized in comparison with the results of traditional market research (e.g., questionnaire survey). Thus, the labor intensity of natural digital information processing and quantification is very high and calls for the development of specialized procedures. In this study, such a procedure was suggested, implemented and automated using the Python programming language for developing reasoned characteristics of context and targeted advertising based on processing the natural digital information generated by prospective consumers of the product advertised on social media. Profound and comprehensive analysis of the results of the applied research made it possible to form a complete picture of the unique methodological properties of market research based on natural digital information processing. The primary methodological property of this type of research is a mechanism for identifying the digital source of this

information. The main selection criterion is the goal setting of the research, according to which the researcher can operate both one or many sources, which can be differentiated into social communication portals, news portals, commercial portals and ideological portals. According to the results of aggregation and primary treatment of natural information, obtained from identified sources, the researcher has to quantify and analyze the key properties of information contained in this array, which, in turn, can be differentiated into content properties, tonality properties, metaproperties and context properties. The properties presented are quantified and analyzed based on specialized metrics. Concrete metrics are determined by the goal setting and specifics of the research. However, four basic classes can be highlighted: connection metrics, comparability metrics, dominance metrics and exceptionality metrics. The analyzed metric results allow the analytical purpose of market research to be achieved. The presented methodological properties are aggregated into a universal algorithmic model for analyzing natural digital information in the context of market research.

Thus, the market impact of our research can be formulated as follows:

- The toolkit developed by the authors (classification of sources of NDI, description of key properties of NDI, metrics to quantify and analyze the properties of NDI and, as a result, the algorithmic model for analyzing natural digital information in the context of market research) allows market research to be conducted without direct attraction of research subjects, which results in cost reduction and elimination of the phenomenon of social desirability.
- The toolkit developed by the authors allows so-called reasoned advertising messages to be created that meet the requests of the target audience, which is proved by the big data that underlie the presented methodology.
- The toolkit developed by the authors is universal for analyzing natural digital information, which, with minor adaptations, can be used by any subject conducting market research.

The results of this study can be of a practical value for internet market researchers, heads of marketing departments, and scholars who conduct research in the field of digital information development, consumer behavior and information environment. The authors suggest that in further research, the proposed algorithmic model should be elaborated in detail, made more specific and saturated with concrete research tools, algorithms and mechanisms for aggregating, quantifying and analyzing natural digital information.

**Author Contributions:** Conceptualization, O.K. and O.Y.; methodology, E.K.; software, E.K.; validation, O.K. and D.R.; formal analysis, O.Y.; investigation, O.K.; resources, D.R.; data curation, O.K.; writing—original draft preparation, O.K. and O.Y.; writing—review and editing, O.K. and O.Y.; visualization, E.K.; supervision, D.R. and O.Y.; project administration, D.R. and O.Y. All authors have read and agreed to the published version of the manuscript.

**Funding:** This research received no external funding.

**Acknowledgments:** The research was supported by the Peter the Great St. Petersburg Polytechnic University.

**Conflicts of Interest:** The authors declare no conflict of interest.

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
