# Peer review of "Analyzing Natural Digital Information in the Context of Market Research"

_information, doi:10.3390/info12100387_

Round 1
Reviewer 1 Report
I enjoyed reading this manuscript as it offers a comprehensive overview of the process involved in using NDI for market research, illustrated with an empirical application. The paper is well written and easy to follow. I only have two very minor suggestions:
1. Please provide further references for the following: "it is commonly known from the perspective of consumer psychology and the theory of 196 market research, that the less a person is aware that he is an object of research, the more 197 reliable is the information that he communicates, even if he does not suspect that."
2. Please clarify: "the stem of the word, which does not necessarily coincide with the root of the word."
Author Response
1. Dear reviewer, thank you very much for your comment about absent reference. We added the reference in the text as well as gave the explanation to the insight about consumer psychology.
2. Also thank you very much for the question about the stemming process. The fact is that the root of a word (especially when it comes to verbs and adjectives) can have a fundamentally different interpretation depending on the prefixes and suffixes. In this regard, stemming emphasizes the stem of the word, and not just the root. We understand that this is a very important clarification, so we added it to the text. Thanks again.
All the revisions we made are marked in the text in red.
Our team of authors would like to express our deep gratitude to you for such a detailed reading of our article and valuable comments, which certainly greatly improve its quality.

Reviewer 2 Report
The paper presents the architecture of natural digital information processing. The authors propose you use this approach for market research. The term of natural digit information is presented. The paper describes the methodology of processing such kind of information. However, this is methodology is similar to Big data preprocessing, especially for semistructured and unstructured data sources.
The following suggestions are given:
1) The abstract is too long. Please use the abstract to underline the impact of your research.
2) What is the difference between natural digital information and Big data? Please underline this in Fig 1.
3) Wich language authors are used in the data processing stage? How different is this stage for different languages?
4) How do authors check the accuracy of the text tonality method?
5) The discussion section should be proved using numerical representation, not only text and authors suggestions.
6) Please demonstrate the market impact of your method according to the paper title.
Author Response
1. Dear reviewer, thank you very much for your comment about the content of our Abstract, we significantly cut it, but at the same time underlined the impact of our research in Abstract.
2. Thank you very much for your question about the difference between natural digital information and Big data. The key difference is that natural digital information is only part of the big data generated by humans in the information environment. It differs from all other information in that it is formulated in a natural form and requires decoding through human consciousness. This is what determines the need for its preliminary quantification with the help of specific tools that we describe. This is really important clarification, therefore, as you recommended, you noted this in Figure 1 and in the text.
3. Dear reviewer, as for your question about which language we used in the data processing stage, the answer is that we analyzed the information in the Russian language. This is due to the fact that the practical case described was implemented in the Russian market. However, our proposed solutions and conclusions are universal for natural digital information presented in any language. Also, it should be noted that many languages have unique specifics, primarily related to the ways and lexical traditions of reflecting this or that information in the text. This specificity should be taken into account, but its presence does not contradict our conclusions and proposed solutions. This moment is really important, so we made the appropriate additions to the text of the article.
4. About your comment on checking the accuracy of the text tonality method, the quality control was carried out selectively. Undoubtedly, the tool we use (the Dostoevsky library for the Python programming language) has a certain inaccuracy, since it is based on a neural network trained on tagged Twitter data. However, its level of accuracy is 71%, which makes it the most accurate available tool for analyzing the tonality of texts presented in Russian. Of course, in the future we plan to develop our own more accurate tonality assessment tool.
5. Thank you very much for your comment about numerical representation of data in the Discussion section. We added numerical representation of metrics to quantify and analyze the properties of Natural digital information in the form of Table 2.
6. Dear reviewer, thank you very much for your comment about clarifying the market impact of our method according to the paper title. Thus, the market impact of our research can be formulated as follows:
- the toolkit developed by the authors (classification of sources of NDI, description of key properties of NDI, metrics to quantify and analyze the properties of NDI and, as a result, the algorithmic model for analyzing natural digital information in the context of market research) allows to conduct market research without direct attraction of research subjects, which results in cost reduction and elimination of the phenomenon of social desirability.
- the toolkit developed by the authors allows to create the so-called reasoned advertising messages that meet the requests of the target audience, which is proved by big data that underlies the presented methodology.
- the toolkit developed by the authors is universal for analyzing natural digital information, which, with minor adaptations, can be used by any subject conducting market research.
This moment is really important, so in the conclusion section, we formulated specific points to demonstrate the market impact of our research.
All the revisions we made are marked in the text in red.
Our team of authors would like to express our deep gratitude to you for such a detailed reading of our article and valuable comments, which certainly greatly improve its quality.

Round 2
Reviewer 2 Report
Authors have taken into account my comments
This manuscript is a resubmission of an earlier submission. The following is a list of the peer review reports and author responses from that submission.